# Trajectories of recall memory as predictive of hearing impairment: A longitudinal cohort study

**Asri Maharani**[1]*, **Piers Dawes**[2], **James Nazroo**[3], **Gindo Tampubolon**[3], **Neil Pendleton**[1], **on behalf of the SENSE-Cog WP1 group**¶

1 Division of Neuroscience and Experimental Psychology, School of Biological Sciences, Faculty of Biology, Medicine and Health, University of Manchester, Manchester, United Kingdom, 2 Division of Human Communication, Development & Hearing, University of Manchester, Manchester, United Kingdom, 3 Cathie Marsh Institute for Social Research, University of Manchester, Manchester, United Kingdom

¶ Membership of the SENSE-Cog WP1 group is provided in the Acknowledgements.
* asri.maharani@manchester.ac.uk

**Data Availability Statement:** The ELSA dataset is freely available from the UK Data Service to all bonafide researchers. The dataset can be accessed

## Abstract

### Objectives

Accumulating evidence points to a relationship between hearing function and cognitive ability in later life. However, the exact mechanisms of this relationship are still unclear. This study aimed to characterise latent cognitive trajectories in recall memory and identify their association with subsequent risk of hearing impairment.

### Methods

We analysed data from the English Longitudinal Study of Ageing Wave 1 (2002/03) until Wave 7 (2014/15). The study population consisted of 3,615 adults aged 50+ who participated in the first wave of the English Longitudinal Study of Ageing, who had no self-reported hearing impairment in Wave 1, and who underwent a hearing test in Wave 7. Respondents were classified as having hearing impairment if they failed to hear tones quieter than 35 dB HL in the better ear.

### Results

The trajectories of recall memory scores were grouped using latent class growth mixture modelling and were related to the presence of hearing impairment in Wave 7. Models estimating 1-class through 5-class recall memory trajectories were compared and the best-fitting models were 4-class trajectories. The different recall memory trajectories represent different starting points and mean of the memory scores. Compared to respondents with the highest recall memory trajectory, other trajectories were increasingly likely to develop later hearing impairment.

### Conclusions

Long-term changes in cognitive ability predict hearing impairment. Further research is required to identify the mechanisms explaining the association between cognitive

here: https://discover.ukdataservice.ac.uk/series/?sn=200011.

**Funding:** SENSE-Cog project has received funding from the Horizon 2020 Framework Programme (633491) research and innovation programme under grant agreement No 668648. PD is supported by the UK National Institute for Health Research (01A1-CSPP04-014). The funders had no role in study design, data collection and analysis, decision to publish, or preparation of the manuscript.

**Competing interests:** The authors have declared that no competing interests exist.

trajectories and hearing impairment, as well as to determine whether intervention for maintenance of cognitive function also give benefit on hearing function among older adults.

## Introduction

Hearing impairment has become a major concern for global health. The 2015 Global Burden of Disease estimates showed that hearing impairment was the fourth leading cause of years lived with disability (YLDs) and those YLDs increase from age 35 years to 64 years [1]. Data from 1989 National Survey of Hearing showed that hearing impairment in both ears was experienced by approximately 21% of adults aged 17–80 years old [2]. A study using data from the English Longitudinal Study of Ageing (ELSA) Wave 2 showed that, in 2004, 39% of UK residents aged 60 years and older reported hearing difficulties and that this proportion increased in more advanced age groups [3].

There is growing evidence that hearing impairment is independently associated with the magnitude of cognitive decline [4, 5] and incident dementia [6–9]. In an earlier analysis of three nationally representative longitudinal data sets from the United States and Europe, we found that sensory (hearing and/or visual) impairment was associated with accelerated cognitive decline among people aged 50 or older and that the association was stronger among those with dual sensory (hearing and visual) impairment [10]. This association was independent of demographic and socioeconomic factors, health behaviours, symptoms of depression, and the presence of chronic diseases.

There is therefore evidence of an association between hearing impairment with cognitive decline and dementia. But the nature of the relationship between hearing impairment and cognitive function is unclear [11, 12]. A number of mechanisms are proposed in the literature: declining cognitive capacity leading to hearing loss, poorer hearing function causing accelerated cognitive decline [6–8], and the presence of third factors causing both types of decline [13, 14]. In relation to the possibility that declining cognitive capacity leads to hearing loss, prior study hypothesises that the declining cognitive ability results in poorer auditory perception as reduced cognitive capacity may constrain perceptual processing of stimuli [12, 15, 16]. Alternatively, hearing loss and cognitive decline may be associated because of a shared common factor(s) [13].

A large amount of experimental research demonstrates that cognitive factors–including memory, attention and processing speed–are critical for 'listening' [17–20]. Cognitive factors also impact on performance on tests of low-level auditory perception, including detection of pure tones [21]. But longitudinal evidence for declining cognitive capacity leading to hearing loss is limited. To our knowledge, only one study has shown that cognitive impairment was associated with faster declines in hearing threshold [22]. That study used data from four waves spanning 11 years of the Mini-Mental State Examination, a verbally administered screen of global cognitive function, to measure the cognitive ability [22]. Unfortunately, the MMSE is sensitive to hearing function and this susceptibility may have confounded the results [23, 24]. In contrast, a study using two waves of data from the Baltimore Longitudinal Study of Aging found that cognitive function in the first wave did not correlate with change in audiometrically assessed hearing function in the second wave [25]. However, those studies used data from two time points only 2 years apart which restrict potential to characterise trajectories of the cognitive function.

We address these issues by employing a three-step latent class analysis with a distal outcome in a population-based cohort study of people with self-reported normal hearing at baseline,

with repeated measures of recall memory available across the older age life course and objective hearing measures in the final wave. This study aims to identify recall memory trajectories in older people living in England and test whether trajectories of recall memory during the preceding decade are associated with the development of hearing impairment.

## Materials and method

This study forms part of the SENSE-Cog multi-phase research programme, funded by the European Union Horizon 2020 programme. SENSE-Cog aims to promote mental well-being in older adults with sensory and cognitive impairments (http://www.sense-cog.eu/). The first work package of this project aims to better understand the links between sensory, cognitive and mental ill-health in older Europeans. Ethical review for this study has been granted by the Ethical Review in H2020 panel number 668468_Sense-cog.

### Subjects

Subjects were participants in the English Longitudinal Study of Ageing (ELSA), an ongoing biannual, nationally representative, longitudinal study of men and women aged 50 and older in England [26]. The ELSA provides information on demographics, socio-economics, social participation, and health. Ethical approval for ELSA was obtained from the National Health Service Research Ethics Committees under the National Research and Ethics Service. All participants gave written informed consent. All methods were performed in accordance with approved guidelines and regulations. The first wave of ELSA (2002/03) recruited 11,391 individuals aged 50 years old and older from households previously responding to the Health Survey for England and their partners. To date, there have been eight waves of ELSA, with data from the first seven waves used here. Information on subjective hearing function is available in all waves, while an objective hearing examination was only performed in Wave 7. The information on hearing function available in the first wave of ELSA included only self-reported hearing quality, which was determined using the question: 'Is your hearing [using a hearing aid as usual] excellent (1), very good (2), good (3), fair (4) or poor (5)?'. We identified participants who reported having fair or poor hearing in the first wave as having a hearing impairment [10]. The present analysis thus includes respondents from Wave 1 aged 50 years and older who did not report a hearing impairment in the first wave and who underwent an objective hearing examination in Wave 7. The final sample consisted of 3,615 individuals. Differences between the analytic sample (3,615) and excluded sample (n = 726) were tested with Kruskal-Wallis one-way analysis of variance for numerical variables and ordinal chi-square tests for categorical variables as appropriate (S1 Table). There were more women than men in the analytic sample, conversely in the excluded sample ($p < 0.001$). The excluded sample had lower recall memory scores on average ($p < 0.001$), lower education attainment ($p < 0.001$), and were poorer (p < 0.001), and older (74 versus 72.2; $p < 0.001$) than the analytic sample. The proportion of the excluded sample with objective hearing impairment in wave 7 (26.8%) was higher than that of analytic sample (6.9%).

### Objective hearing function measure

Objective hearing function in respondents to Wave 7 was measured using a hearing screening device (HearCheck Screener, Siemens, Germany). The Health Survey for England validated this device in 2014 [27]. Respondents reported how many of the three tones they heard for each frequency (1000 Hz and 3000 Hz) in each ear. Respondents were categorised as not having hearing impairment if they heard at least two low-pitched sounds (1000 Hz) and at least one high-pitched sound (3000 Hz) in the better ear, corresponding to audiometric hearing

thresholds greater than 35 dB HL at 1000 Hz and 3000 Hz. These thresholds have been identified as being the most useful frequencies for screening for hearing impairment, indicating a level of hearing impairment such that a person would likely benefit from a hearing aid [27]. For the sensitivity analysis, we used the sum of the number of tones (both low- and high-pitched sounds) that participants heard in both ears. Objective hearing function (as measured by the number of tones that participants heard in both ears) at wave 7 had a significant correlation with subjective hearing function (indicated on the 5-point Likert scale described above) at the same wave (Spearman correlation = -0.47, $p<0.001$).

## Recall memory measure

We used recall memory scores to measure cognitive function. The recall memory test consisted of verbal recall of a list of ten simple words. Respondents heard the complete list only once, and the test was carried out twice: immediately after the words were read out (immediate recall) and at the end of the cognitive function module (delayed recall). The raw total scores of both tests correspond to the number of words that the respondent recalled. The maximum recall memory score is 20.

## Confounders

We included an extensive set of confounders, measured at Wave 7, that are known to affect hearing function among older adults [3, 28]. Demographic covariates included age, sex (with male as the reference) and marital status (married/partnered, divorced, widowed, and single as the reference). Socioeconomic determinants include, education (primary as the reference, high school, and college or higher) and wealth tertiles (poorest as the reference, middle, and richest). We used the aggregate of private pension wealth and state pension wealth to measure the wealth of the respondents.

Hearing function is known to be affected by health behaviour and health status [29, 30]. The measures of health behaviour included were smoking, alcohol consumption and physical activity level. Respondents were categorised as current smokers, past smokers, or non-smokers. For alcohol consumption we classified respondents as drinking regularly if they consumed alcohol 5–7 days per week. We categorised respondents as engaging in moderate and vigorous physical exercise if they reported exercising as least once a week. The respondents responded to the questions whether they take part in sports or activities that are vigorous and moderately energetic, respectively. For health status, we included a series of indicators of chronic conditions, based on positive medical history (self-report of "has been diagnosed by doctors"), including diabetes, heart attack, hypertension, chronic lung disease, stroke, and cancer.

## Statistical analysis

This study comprised three analytic phases of latent class modelling with distal outcomes to identify the relationship between various cognitive trajectories and hearing impairment [31]. In the first phase, we built latent class trajectory models to identify trajectories of recall memory scores over 13 years. Latent class trajectory model is an extension form of finite mixture modelling, which describes the course of recall memory scores through a regression function using continuous latent growth factors [32]. We used a latent class trajectory model as it could classify respondents into distinct groups based on their response patterns so that respondents within a group are more similar than respondents between groups. The intercept represents the level of recall memory scores at baseline. The change in recall memory scores over time is accounted for by the linear and quadratic slopes of the growth factors. We calculated the

posterior probabilities for each trajectory taking into account the respondents' age (at baseline), sex, and educational level.

We then assigned the respondents post hoc to the trajectory with highest probability in the second phase. We selected the best model fit among a finite set of the models including up to five classes based on the following criteria: lower Bayesian Information Criteria (BIC), every class contains more than 5% of the respondents, and distinction between classes. S2 Table shows the regression parameters of the latent class models, while S3 Table provides information on the fit criteria for each set of models. The characteristics of respondents at wave 7 (sociodemographic, health behaviour, and health status) were compared across latent class trajectories of recall memory scores (highest, 2nd, 3rd, and lowest) using ordinal chi-square tests for categorical variables and Kruskal-Wallis one-way analysis of variance for numerical variables.

In the third phase, a logistic regression of hearing impairment on recall memory trajectories while controlling for other determinants in wave 7 was explored. The previously developed four-class trajectories were maintained. Using the model-based latent class analysis approach to distal outcomes incorporates hearing impairment as a continuous external consequence and produces coefficients expressing the probability of having hearing impairment given latent class membership while keeping the classification errors [31]. For this second analysis, hearing impairment was defined as the inability to hear tones at 35 dB HL at 1000 Hz and 3000 Hz in the better ear. We conducted latent class modelling with a distal outcome, with hearing function (the sum of the number of tones participants heard in both ears) as a continuous external consequence for the sensitivity analysis. We used sampling weights for all analyses to adjust for non-response and to ensure population representativeness. The analyses were conducted using LatentGOLD 5.1 and STATA 16.

## Results

The Wave 7 (2014–2015) sample profile is described in Table 1. The study sample comprised 3,615 respondents (59.7% female) with a mean age of 72.2 years (see Table 1, second column from the left). The mean episodic memory score was 10.2. Most of the respondents were married (61.9%) and had finished college or higher education (53.4%). Approximately 7.9% and 21.0% of respondents were current smokers and drank alcohol regularly, respectively. Slightly more than 60% of respondents engaged in moderate physical activity, while only 17.7% of respondents engaged in vigorous physical activity at least once a week.

After examining fit statistics, latent class prevalence, and interpretability, we found four trajectories of memory change over the 13-year period under study: the highest (12.2% of the sample), 2nd highest (38.6%), 3rd highest (38.4%), and lowest recall memory trajectories (10.6%) (see Fig 1). The recall memory trajectories in Fig 1 differ in the intercepts and mean of the memory scores. The trajectory of recall memory among the respondents in the highest recall memory group ends at 86 as they are younger than other age groups. The oldest respondent in that group was aged 86 in the final wave. The four middle columns in Table 1 summarise the characteristics of respondents according to trajectory class in Wave 7. Respondents with the highest recall memory trajectory were likely to perform better on the memory test over time, younger, to be wealthier, and better educated than those with lower recall memory trajectories. Furthermore, the highest proportion of respondents engaging in moderate or vigorous physical activity at least once a week was found in the group with the highest recall memory trajectory.

The results of putting together contemporaneous risk factors of hearing impairment are presented in Table 2, left column. We refined this initial model by adding indicators of recent

**Table 1. Descriptive statistics of the analytic sample in Wave 7 (n = 3,615).**

| Variable | All* | Highest recall memory* | 2nd * | 3rd * | Lowest recall memory | p-value |
|---|---|---|---|---|---|---|
| Episodic memory score | 10.2 (3.6) | 14.9 (2.1) | 11.8 (2.1) | 8.7 (2.4) | 4.1 (2.3) | <0.001 |
| Age | 72.2 (6.9) | 67.6 (4.7) | 70.3 (5.8) | 74.3 (6.9) | 78.4 (6.7) | <0.001 |
| Female | 2,158 (59.7) | 322 (72.6) | 852 (60.9) | 765 (55.0) | 219 (56.8) | <0.001 |
| *Marital status* | | | | | | <0.001 |
| Single | 165 (4.5) | 20 (4.5) | 67 (4.8) | 52 (3.7) | 26 (6.7) | |
| Married | 2,241 (61.9) | 312 (70.4) | 956 (68.4) | 810 (58.2) | 163 (42.3) | |
| Divorced | 425 (11.7) | 52 (11.7) | 171 (12.2) | 168 (12) | 34 (8.8) | |
| Widowed | 784 (21.6) | 59 (13.3) | 203 (14.5) | 360 (25.9) | 162 (42.0) | |
| *Education* | | | | | | <0.001 |
| Primary | 963 (26.6) | 16 (3.6) | 208 (14.8) | 497 (35.7) | 242 (62.8) | |
| High school | 722 (19.9) | 85 (19.1) | 336 (24.0) | 263 (18.9) | 38 (9.8) | |
| College or higher | 1,930 (53.3) | 342 (77.2) | 853 (61.0) | 630 (45.3) | 105 (27.2) | |
| *Wealth* | | | | | | <0.001 |
| 1st tertile (poorest) | 1,096 (30.3) | 71 (16.0) | 346 (24.7) | 475 (34.1) | 204 (52.9) | |
| 2nd tertile | 1,260 (34.8) | 123 (27.7) | 485 (34.7) | 518 (37.2) | 134 (34.8) | |
| 3rd tertile (richest) | 1,259 (34.8) | 249 (56.2) | 566 (40.5) | 397 (28.5) | 47 (12.2) | |
| *Smoking Behaviour* | | | | | | 0.001 |
| Non-smoker | 1,547 (42.7) | 221 (49.8) | 611 (43.7) | 572 (41.1) | 143 (37.1) | |
| Past smoker | 1,782 (49.2) | 195 (44.0) | 677 (48.4) | 710 (51.0) | 200 (51.9) | |
| Current smoker | 286 (7.9) | 27 (6.0) | 109 (7.8) | 108 (7.7) | 42 (10.9) | |
| *Drinking Behaviour* | | | | | | |
| Drinking regularly | 681 (21.0) | 123 (29.0) | 280 (21.7) | 245 (19.6) | 33 (12.0) | <0.001 |
| *Physical activity* | | | | | | |
| Moderate physical activity | 2,197 (60.7) | 336 (75.8) | 982 (70.3) | 763 (54.8) | 116 (30.1) | <0.001 |
| Vigorous physical activity | 642 (17.7) | 117 (26.4) | 306 (21.9) | 197 (14.1) | 22 (5.7) | <0.001 |
| *The presence of chronic diseases* | | | | | | |
| Heart diseases | 223 (6.1) | 12 (2.7) | 64 (4.5) | 100 (7.1) | 47 (12.2) | <0.001 |
| Diabetes mellitus | 475 (13.1) | 31 (7.0) | 165 (11.8) | 201 (14.4) | 78 (20.6) | <0.001 |
| Stroke | 207 (5.7) | 9 (2.0) | 58 (4.1) | 91 (6.5) | 49 (12.7) | <0.001 |
| Cancer | 247 (6.8) | 23 (5.1) | 91 (6.5) | 101 (7.2) | 32 (8.3) | 0.280 |
| Lung diseases | 202 (5.5) | 14 (3.1) | 65 (4.6) | 98 (7.0) | 25 (6.4) | 0.004 |

* Presented are mean (SD) or number (%).

recall memory trajectories to arrive at the final model (right column). In the initial model, several risk factors show strong and significant associations with hearing impairment. Age had a significant and positive relationship with the presence of hearing impairment. Being female, having higher education, being wealthy, drinking alcohol regularly and being more active were correlated with lower odds of having hearing impairment.

When the 13-year-long cognitive trajectories were added to the initial model, they proved to be significant (right column). The highest recall memory trajectory served as the referent. The relationships between cognitive trajectories and hearing impairment showed a graded effect. Relative to those with most advantaged trajectory, those with the 2nd highest, 3rd highest, and lowest recall memory trajectories were more than twice, four and seven times more likely than the reference (highest memory scores) to suffer hearing impairment, respectively. The grading effect also appears in the sensitivity analysis (see S4 Table). Those with the 2nd highest ($\beta$ = -0.17, $p<0.001$), 3rd highest ($\beta$ = -0.47, $p<0.001$), and lowest recall memory

trajectories (β = -1.70, $p<0.001$) were able to hear fewer tones in the hearing test. This sensitivity analysis suggests that the results are robust to the measure of hearing impairment used.

## Discussion

In a population-based sample of older English adults, we identified four trajectories of recall memory change over a 13-year period (see Fig 1). Supporting prior research [33], we identified the highest recall memory trajectory and the lowest recall memory trajectory over time. We further demonstrated that those four recall memory trajectories strongly predict hearing impairment. Not only is an individual's past trajectory of recall memory predictive of hearing loss, it is also the strongest predictor by far. For example, members of the group with the lowest recall memory trajectory have seven times higher odds to have hearing impairment at the end of more than a decade of study than members of the group with the highest recall memory trajectory.

### Mechanisms and implications

From a theoretical perspective, this study provides the first evidence that decreased recall memory ability predicts hearing impairment. The pathways between cognition and hearing

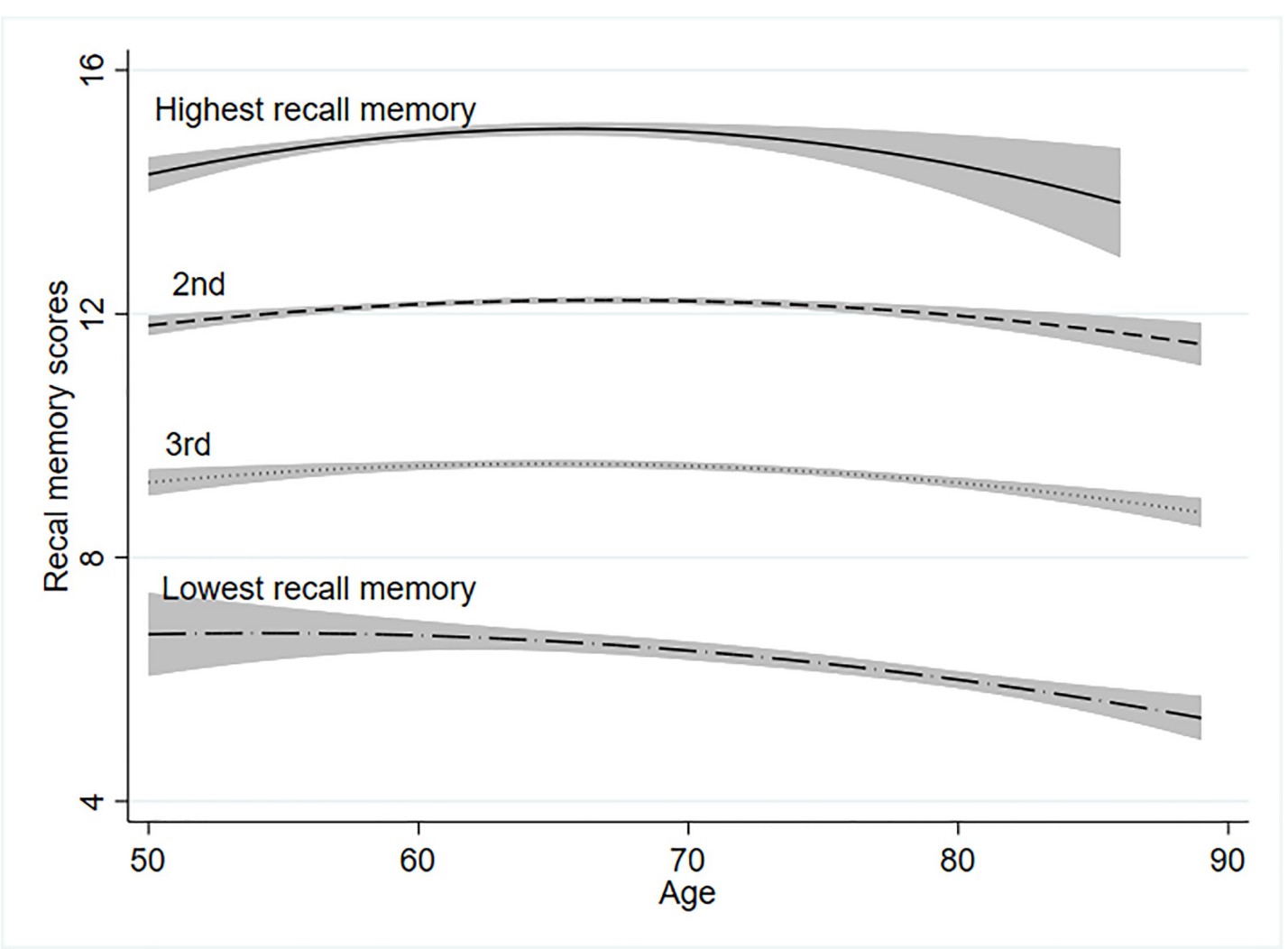

**Fig 1. Four trajectories of episodic memory scores.** Source: ELSA Waves 1–7 (2002–2015).

**Table 2. Results for hearing impairment models (with and without recall memory trajectories).**

| | Model 1 (without recall memory trajectories) * | Model 2 (with recall memory trajectories) * |
|---|---|---|
| *Class trajectories (Ref: highest memory)* | | |
| 2nd | | 2.38 (1.99; 2.84) ‡ |
| 3rd | | 4.63 (3.73; 5.74) ‡ |
| Lowest memory | | 7.9 (5.42; 11.53) ‡ |
| Age | 1.01 (1.00; 1.01) † | 1.00 (1.00; 1.01) † |
| Female | 0.64 (0.52; 0.73) ‡ | 0.79 (0.70; 0.90) ‡ |
| *Education (Ref: Primary)* | | |
| High school | 0.61 (0.51; 0.72) ‡ | 0.89 (0.74; 1.07) |
| College or higher | 0.74 (0.65; 0.85) ‡ | 0.88 (0.76; 1.01) |
| *Marital status (Ref: Single)* | | |
| Married | 1.16 (0.83; 1.61) | 1.23 (0.88; 1.71) |
| Divorced | 1.37 (0.95; 1.98) | 1.51 (1.04; 2.19) † |
| Widowed | 2.48 (1.75; 3.5) ‡ | 2.25 (1.58; 3.13) ‡ |
| *Wealth (Ref: 1$^{st}$ tertile)* | | |
| 2$^{nd}$ tertile | 1.01 (0.88; 1.15) | 1.11 (0.96; 1.27) |
| 3$^{rd}$ tertile (richest) | 0.58 (0.49; 0.68) ‡ | 0.73 (0.61; 0.87) ‡ |
| *Smoking behaviour (Ref: Non-smoker)* | | |
| Past smoker | 1.49 (1.32; 1.69) ‡ | 1.48 (1.31; 1.67) ‡ |
| Current smoker | 0.85 (0.66; 1.09) | 0.85 (0.65; 1.1) |
| Drinking regularly | 0.70 (0.6; 0.83) ‡ | 0.75 (0.63; 0.89) ‡ |
| Moderate physical activity | 0.71 (0.63; 0.8) ‡ | 0.86 (0.76; 0.97) † |
| Vigorous physical activity | 0.79 (0.66; 0.96) | 0.88 (0.73; 1.06) |
| *The presence of chronic diseases* | | |
| Heart diseases | 1.17 (0.95; 1.44) | 1.11 (0.90; 1.37) |
| Diabetes mellitus | 1.17 (1; 1.36) † | 1.14 (0.98; 1.33) |
| Stroke | 1.35 (1.09; 1.66) ‡ | 1.24 (1; 1.53) † |
| Cancer | 0.84 (0.67; 1.06) | 0.83 (0.65; 1.05) |
| Lung diseases | 0.94 (0.75; 1.19) | 1.00 (0.79; 1.27) |

* Presented are odds ratios (confidence intervals).

Sig.

†: significant at 5% or less

‡: significant at 1% or less.

function among older adults are not well understood. One plausible mechanism of the relationship is neurobiological. Research has established the positive correlations between cognitive function and total brain volume [34–36], which may influence the functioning of auditory processes in the brain. Histopathological involvement of auditory cortices has been found in Alzheimer's disease and other dementias [37–39]. The general process in the central auditory system includes matching new inputs with existing memory as well as experience of the auditory world with other sensory systems [40, 41]. In addition to the process within the auditory system, there is a cortico-cortical process that provides integration of the auditory process and other sensory systems, memory, knowledge, and decision-making processes. The neurodegenerative process may disrupt both central auditory and cortico-cortical processes, leading to deficient perception of sounds and a more cautious or impaired decision-making process. A

study in the US found that auditory cortical activity as measured by P50 amplitude and P300 latency decreased with normal ageing and decreased additionally with mild cognitive impairment [42].

Another potentially relevant mechanism is a shared underlying pathology, such as vascular diseases or intrinsic cellular ageing, that leads to ageing of both the brain and the auditory organs [13, 14, 16, 43, 44]. However, further explanation is warranted because the association between recall memory trajectories and the presence of hearing impairment in the present analyses remained after controlling for the effects of potential covariates (e.g. age, stroke, heart diseases).

Finally, recent trajectories of recall memory scores could also potentially predict hearing impairment through the broader influence of cognitive ability on physical activity. Likewise, social networks could mediate the observed association of cognition and a higher probability of hearing impairment. Better cognition may facilitate physical activity [45], which in turn protects older adults against chronic diseases [46, 47]. The presence of chronic diseases, including stroke, diabetes, hypertension and other cardiovascular diseases, are among the risk factors for hearing impairment [28–30]. Better cognitive function is associated with richer social networks [48], and similar to physical activity, social relationships are important for good health [49, 50].

From a clinical perspective, the findings from the current study emphasise the importance of assessing auditory function for patients with cognitive impairment or dementia to identify those who are at increased risk for future hearing impairment. Hearing loss is highly prevalent [1–3], and hearing loss may be both preventable and treatable with rehabilitative devices and strategies that remain mostly underutilised. As the timing of rehabilitation is a crucial factor for the success of interventions in old age, early identification of hearing loss may increase the success rates of treatment and reduce the impact of hearing loss. Our findings that cognitive decline predicts future hearing impairment may also present a challenge for adjustment to new hearing aids and effective use of hearing aids among those with cognitive limitations.

Another suggestion derivable from our findings is that older adults with hearing impairments might profit from interventions designed to prevent cognitive decline, such as physical activity [51] and the multidomain intervention (combination between diet, exercise, cognitive training and vascular risk monitoring) [52, 53]. Well-designed future studies are needed to determine whether those interventions can attenuate or prevent hearing impairment.

This study has several limitations. Firstly, there were no objective data on hearing at baseline. Self-reported hearing a good indicator of hearing status, but tends to under-identify cases of hearing impairment [54]; so people with a hearing impairment may have been included at baseline. The second limitation is that the present study was observational, so we were not able to interpret the association between cognitive trajectories and hearing impairment as causal. Future randomised controlled trials are required to confirm causality. Although we controlled for a wide range of potential confounders, other unmeasured factors could be important. Thirdly, the objective measures of hearing function involved the HearCheck audiometric screener device instead of full audiometry. The proportion of the respondents with objective hearing impairment included in this study was lower for the age group (6.9%). Prior studies using national samples showed that the prevalence of hearing impairment among adults aged 50 years and older ranges from 20 to 40% [55]. The definition of hearing loss in this study was not the usual four frequency average of thresholds at 500, 1000, 2000 and 4000 Hz. It was performance on a pure tone audiometry screen with hearing loss equivalent to better ear threshold >35 dB HL at 1000 or 3000 Hz. Hearing loss usually appears first at higher frequencies, so including lower frequencies in an average hearing loss may make the hearing loss index less sensitive to early hearing loss. The 3000 Hz criterion for our hearing loss measure should

make it reasonably sensitive to early hearing loss. The accuracy of HearCheck has been established in a sample of adults a similar age to those in the cohort in this study [56]. HearCheck had 89% sensitivity and 62% specificity with respect of audiometrically tested hearing at 35 dB HL. Fourth, the cognitive ability was measured using verbally administered tests. The relationship between the trajectories of recall memory and hearing function in this study may therefore be an artefact of verbal assessment. Finally, episodic memory as the main outcome does not define cognitive ageing; it is only one among many cognitive functions that change with age [57]. Furthermore, the age-related changes in those different cognitive abilities occur at different rates [58, 59]. Episodic memory, however, does represent one of the earliest indicators of neurodegenerative disorders [60] and is important for financial decision-making in later life [61, 62].

## Conclusions

In conclusion, the results of this study point to a new finding in cognitive ageing: that 13-year cognitive trajectories predict hearing status. Our results add new evidence linking cognitive ability to hearing function, suggesting the importance of addressing cognitive abilities when assessing hearing performance and, conversely, of being aware of hearing impairment when diagnosing, screening and managing patients with cognitive impairment or dementia. We further suggest that interventions aimed at improving cognition (e.g. physical activity (51) and its combination with other domain [52, 53] may also bring benefit on hearing performance.

## Supporting information

**S1 Table. Comparisons of the excluded and analytic sample in Wave 7.**
(DOCX)

**S2 Table. Regression parameters of the latent class models.**
(DOCX)

**S3 Table. Fit criteria for set of latent class models.**
(DOCX)

**S4 Table. Results for hearing function models (with and without recent trajectories of recall memory).**
(DOCX)

## Acknowledgments

The Sense-Cog WP1 group are Geir Bertelsen[1,2], Suzanne Cosh[3], Audrey Cougnard-Grégoire[4], Piers Dawes[5], Cécile Delcourt[4], Fofi Constantinidou[6], Andre Goedegebure[7], Catherine Helmer[4], M. Arfan Ikram[8,9], Caroline CW Klaver[8,10], Iracema Leroi[11], Asri Maharani[11,12], Magda Meester-Smor[8,10], Virginie Nael[4], Neelke Oosterloo[7]; Neil Pendleton[11,12], Henrik Schirmer[13], Gindo Tampubolon[14], Henning Tiemeier[8,15], Therese von Hanno[16,17].

1. UiT The Arctic University of Norway, Department of Community Medicine, Faculty of Health Sciences, N-9037 Tromsø, Norway

2. University Hospital of North Norway, Department of Ophthalmology, N-9038 Tromsø, Norway

3. School of Psychology, University of New England, Armidale, Australia

4. Univ. Bordeaux, Inserm, Bordeaux Population Health Research Center, team LEHA, UMR 1219, F-33000 Bordeaux, France

5. University of Manchester, Manchester Centre for Audiology and Deafness, School of Health Sciences, Manchester, UK

6. University of Cyprus, Department of Psychology & Center for Applied Neuroscience, Nicosia, Cyprus

7. Erasmus Medical Centre, Department of Otorhinolaryngology and Head and Neck Surgery, Rotterdam, the Netherlands.

8. Erasmus Medical Centre, Department of Epidemiology, Rotterdam, The Netherlands

9. Erasmus Medical Centre, Departments of Neurology and Radiology, Rotterdam, The Netherlands

10. Erasmus Medical Centre, Department of Ophthalmology, Rotterdam, The Netherlands

11. University of Manchester, Division of Neuroscience and Experimental Psychology, School of Biological Sciences, Manchester, UK

12. University of Manchester, Academic Health Science Centre, Manchester, UK

13. UiT-The Arctic University of Norway, Department of Clinical Medicine, Cardiovascular research Group-UNN, N-9037 Tromsø, Norway

14. University of Manchester, Global Development Institute, Manchester, UK

15. Erasmus Medical Centre, Department of Psychiatry, Rotterdam, The Netherlands

16. UiT-The Arctic University of Norway, Department of Clinical Medicine, Faculty of Health Sciences, N-9037 Tromsø, Norway

17. Nordland Hospital, Department of Ophthalmology, N-8092 Bodø, Norway

## Author Contributions

**Conceptualization:** Asri Maharani, Piers Dawes, James Nazroo, Gindo Tampubolon, Neil Pendleton.

**Data curation:** Asri Maharani.

**Formal analysis:** Asri Maharani.

**Funding acquisition:** Piers Dawes, James Nazroo, Neil Pendleton.

**Investigation:** Asri Maharani.

**Methodology:** Asri Maharani, Piers Dawes, James Nazroo, Gindo Tampubolon, Neil Pendleton.

**Resources:** Asri Maharani.

**Software:** Asri Maharani.

**Writing – original draft:** Asri Maharani.

**Writing – review & editing:** Asri Maharani, Piers Dawes, James Nazroo, Gindo Tampubolon, Neil Pendleton.

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
