## [Decision Letter · Decision Letter 0]

5 Dec 2019

PONE-D-19-29591

Trajectories of cognitive function as predictive of hearing impairment: A longitudinal cohort study

PLOS ONE

Dear Dr Maharani,

Thank you for submitting your manuscript to PLOS ONE. After careful consideration, we feel that it has merit but does not fully meet PLOS ONE’s publication criteria as it currently stands. Therefore, we invite you to submit a revised version of the manuscript that addresses the points raised during the review process.

The manuscript deals with an important research question of the mechanisms behind the decline of cognitive functioning and sensory impairments.

It seems relevant to allude a bit more to the biological and other mechanisms behind cognitive and sensory decline already in the introduction.

Conceptual issues:

Since there are studies available that have investigated the simultaneous change of hearing ability and recall memory, the main point to address in a revision is a stronger motivation of the study’s research question. Weak points of the study as it stands now are, firstly, the non-availability of objective hearing measurements before wave 7. Secondly, as the sample as it is set up is quite selective – limiting to respondents who participate in several waves, excluding those with subjective hearing impairment at first wave – one needs to include weights to speak of a nationally representative sample. Third, I am not fully convinced by the idea to use cognitive tests to predict hearing impairment. There is an individual and societal impact of hearing impairment in terms of years lived with disability, but the impact of memory problems on years lived with disability, plus associated social and healthcare costs are much higher for memory problems. Lastly, there is at the moment little hope to be able to dramatically change the course of cognitive trajectories soon – the FINGER intervention only published their results for the two-year follow-up so far, and aside from this study there is little evidence that cognitive decline can be postponed or even reversed. Further, we don’t know anything about the drug that was recently submitted for FDA approval to work against Alzheimer’s at the moment, so a medical cure is also not in immediate sight.

Methodological issues:

- I agree with the methodological issues pointed out by both reviewers. I would like to ask you address them by carrying out and reporting the suggested additional analyses.

- It seems reasonable to select people without subjective hearing impairment at t1, but I suggest to include more information on the differences between this subjective measure (asking about hearing impairment with usual aids, i.e. compensating possibly existing hearing impairment) and an objective hearing test.

- What is the correlation between subjective hearing impairment at wave 7 and the objective hearing test?

- The claim to predict hearing impairment based on the cognitive trajectories more than a decade in advance needs to be substantiated by a test on the ability of the intercept of these trajectories, i.e. a time point more than a decade earlier than the hearing test, to predict hearing impairment.

The revision, as pointed out above, will need to address all of the above-mentioned points, particularly present a stronger motivation and the suggested additional analyses by the reviewers.

We would appreciate receiving your revised manuscript by Jan 19 2020 11:59PM. To enhance the reproducibility of your results, we recommend that if applicable you deposit your laboratory protocols in protocols.io, where a protocol can be assigned its own identifier (DOI) such that it can be cited independently in the future. For instructions see: http://journals.plos.org/plosone/s/submission-guidelines#loc-laboratory-protocols

We look forward to receiving your revised manuscript.

Kind regards,

Anja K Leist, Dr. rer. nat. (equiv. PhD)

Academic Editor

PLOS ONE

Journal Requirements:

This work was supported by SENSE-Cog project. This project has received funding from the European Union’s Horizon 2020 research and innovation programme under grant agreement No 668648.

SENSE-Cog project has received funding from the Horizon 2020 Framework Programme (633491) research and innovation programme under grant agreement No 668648. PD is supported by the UK National Institute for Health Research (01A1-CSPP04-014).

3.  One of the noted authors is a group or consortium: SENSE-Cog WP1 group

In addition to naming the author group, please list the individual authors and affiliations within this group in the acknowledgments section of your manuscript. Please also indicate clearly a lead author for this group along with a contact email address.

4. Please provide additional details regarding participant consent. In the ethics statement in the Methods and online submission information, please ensure that you have specified whether consent was written or verbal/oral. If consent was verbal/oral, please specify: 1) whether the ethics committee approved the verbal/oral consent procedure, 2) why written consent could not be obtained, and 3) how verbal/oral consent was recorded. If your study included minors, please state whether you obtained consent from parents or guardians in these cases.

Reviewers' comments:

Reviewer's Responses to Questions

**Comments to the Author**

1. Is the manuscript technically sound, and do the data support the conclusions?

Reviewer #1: Yes

Reviewer #2: No

2. Has the statistical analysis been performed appropriately and rigorously? 

Reviewer #1: Yes

Reviewer #2: No

3. Have the authors made all data underlying the findings in their manuscript fully available?

Reviewer #1: Yes

Reviewer #2: No

4. Is the manuscript presented in an intelligible fashion and written in standard English?

Reviewer #1: Yes

Reviewer #2: Yes

5. Review Comments to the Author

Reviewer #1: This article adds to the growing literature teasing out hearing-cognition relations in later adulthood. Unlike the majority of research in this area which focuses on hearing as a risk-factor for future cognitive decline – this study considers the opposite direction and explores cognitive decline as a predictor of future hearing impairment. This is its key contribution to the literature.

The article is clearly written, the research question is justified with good rationale, and the methods relatively sound. I have the following comments to make:

Introduction:

Page 3, line 68: As I understand it, Kiely et al did include time-dependent measure of cognition and found incidence of cognitive impairment to be linked with higher hearing thresholds. So , yet limitations of Kiely et al remain – they used a verbally administered screen of global cognitive function (the MMSE) and so were unable to look at cognitive domains or actuals trajectories. Thus, the work of this paper provides important addition .

One other important study to include is Armstrong et al 2018 Journals of Gerontology ( https://doi.org/10.1093/gerona/gly268) who used cross-lagged panel regression to explore reciprocal relationship between hearing and cognition over two time points. They did not find prior cognition to predict future hearing levels. I also encourage authors to consider the findings of Lindenberger and Ghisletta 2009 Psych and Aging (https://psycnet.apa.org/record/2009-03151-001) and the earlier work of Baltes and Lindenberger (‘94, ‘97) who laid the theoretical foundations for this research question.

Methods

A self rated hearing item has been applied to select the sample. This is appropriate as it restricts the sample to those more likely to be later identified with hearing impairment due to age related processes.

The definition of audiometric defined hearing impairment used at wave 7 is also reasonable, but should be made clear that this differs from the more commonly applied PTA>25dB (mild), PTA>40dB (moderate) for 500-4000 Hz etc…

Cognitive measures appear to be verbally administered – associations between hearing and cognition may therefore be (in part) an artefact of verbal assessment. This is a common limitation in this line of research but should be acknowledged in the discussion.

Much more information on the modelling of the latent classes is needed. Are the latent classes derived from intercept and slope factors from a latent growth curve? How was variance within the classes estimated? etc...

Table 1

Without referring to deep into the results section it is not clear what is meant by “most advantaged”. What is the variable stratifying the sample in the columns? This should be made obvious in the table header.

Please ensure consistent reporting to decimal places. (e.g. age SD reported as 7 rather than 7.0 for class 3 and class 4)

Table 1 refers to wealth. Is this wealth or income? In text it appears to be referred to as income.

Figure 1 and the latent classes

These are interesting, but appear to simply reflect differences in levels? Are there differences in the way these classes are changing over time?

Why does the line in figures 1 for most advantaged class stop around age 85 whereas other classes lines continue to age 90? It looks like the grades of cognitive advantaged trajectories are strongly tied with age. Is this true? I would like to see mention of this on page 9-10 (lines 180-184) where a description of characteristics of each class is proved. i.e. most advantaged class has higher income, more education, more social engagement, higher levels of exercise AND is also younger, which may explain all these differences?

I wonder if it would have been simpler, and just as instructive, to regress hearing impairment at wave 7 on latent factors for intercept and slope that are estimated from a latent growth curve for cognition.

Table 2

Am I correct in interpreting the coefficients as the log-odds as the outcome is binary? Is this effectively a logistic regression? Some readers might benefit from reporting of odds ratios.

Discussion:

Overall the discussion is very nicely presented.

Again, I felt the manuscript would benefit from bringing stronger theoretical foundations earlier outlined by Baltes and Lindenberger. Schneider and Pichora-Fuller’s chapter in Craik and Salthouse’ Handbook of Aging and Cognition (2000) also give very thorough account of this.

There is discussion of clinical perspectives and managing hearing loss. What is missing (or could at least be made more clearly) is consideration of the implications of these findings for hearing aid use. If cognitive impairment/decline/problems predict future hearing impairment, then this may presents a challenge for adjustment to new hearing aids and effective use of hearing aids among those with cognitive limitations.

Reviewer #2: This study evaluates associations between classes of recall memory trajectories and hearing impairment in old age. The classes were derived from latent class growth mixture models fitted to wave 1-7 from the English Longitudinal Study of Aging, and hearing impairment was defined as the inability to hear tones quieter than 35 dH HL at wave 7.

Given the way the data analyses were executed, I found it difficult to grasp what is the actual contribution of the study beyond what we already know from previous work, i.e., that memory performance and hearing impairment are correlated (at the population level) in old age. It is unclear what combination of parameters in the growth models are used to derive and define the classes, is it the level, slope, variance/covariance structure, residuals, and so forth? This information is essential for interpretation of the findings.

Also, reasons and justification for the model selection is unclear to my. What is actually gained by relying on latent class modelling in the context of the data and research questions? Is a multivariate normal assumption unrealistic for some reasons? Are there any neuronal factors or data patterns that may justify the mixture assumption? To me, it seems more informative to rely on the multivariate normal assumption and simply fit a conventional growth curve model to the data using the hearing impairment as covariate determining recall level and change, unless there are some specific reasons why that model could be misleading.

For matter of communicative precision I suggest that you replace “cognitive function” for “recall memory” in the title (and the rest of the manuscript).

6. PLOS authors have the option to publish the peer review history of their article (what does this mean?). If published, this will include your full peer review and any attached files.

Reviewer #1: No

Reviewer #2: No

---

## [Author Response · Author response to Decision Letter 0]

13 Feb 2020

Dear Reviewer,

We are very pleased with the review and your efforts in giving feedback on the manuscript. Your recommendations were very useful to strengthen the article. Below you will find your comments numbered and in italic, each followed by a short description of how the comment is implemented in the revised manuscript (in normal font). We have highlighted the changes within the revised manuscripts using coloured text. The number of page and paragraph in our responses refer to the revised manuscripts. We hope that we succeeded in strengthening the paper.

Yours sincerely,

The authors

Editor:

Conceptual issues:

Since there are studies available that have investigated the simultaneous change of hearing ability and recall memory, the main point to address in a revision is a stronger motivation of the study’s research question. 

Weak points of the study as it stands now are, firstly, the non-availability of objective hearing measurements before wave 7. 

Thank you for the comments. We have included the non-availability of objective hearing measurements before wave 7 in the limitation section in Discussion (page 17, lines 309-312):

“Firstly, there were no objective data on hearing at baseline. Self-reported hearing a good indicator of hearing status, but tends to under-identify cases of hearing impairment [54]; so people with a hearing impairment may have been included at baseline.”

Secondly, as the sample as it is set up is quite selective – limiting to respondents who participate in several waves, excluding those with subjective hearing impairment at first wave – one needs to include weights to speak of a nationally representative sample. 

We have included weight in the analyses and included the explanation on it in page 9, lines 196-197:

“We used sampling weights for all analyses to adjust for non-response and to ensure population representativeness.”

Third, I am not fully convinced by the idea to use cognitive tests to predict hearing impairment. There is an individual and societal impact of hearing impairment in terms of years lived with disability, but the impact of memory problems on years lived with disability, plus associated social and healthcare costs are much higher for memory problems.

We have included the literature offering explanation on how cognitive tests predict hearing impairment in the Introduction section (pages 3-4, lines 67-71):

“Prior studies has presented some hypotheses to explain the first mechanism, including that the declining cognitive ability results in poorer perception as it places cognitive load on perception [12, 15, 16]. Another hypothesis is the presence of a common cause factor related to ageing underlying both cognitive impairment and poor hearing function [13].”

We further point out the potential in looking beyond simply cognitive function when assessing older adults’ limitation. A potential of cognitive test predicting future hearing impairment is that one might propose an assessment which considering future potential to have hearing impairment for older adults who report memory problems or have cognitive impairment (page 16, lines 294-296):

“From a clinical perspective, the findings from the current study emphasise the importance of assessing auditory function for patients with cognitive impairment or dementia to identify those who are at increased risk for future hearing impairment.”

Lastly, there is at the moment little hope to be able to dramatically change the course of cognitive trajectories soon – the FINGER intervention only published their results for the two-year follow-up so far, and aside from this study there is little evidence that cognitive decline can be postponed or even reversed. Further, we don’t know anything about the drug that was recently submitted for FDA approval to work against Alzheimer’s at the moment, so a medical cure is also not in immediate sight.

We have included the studies looking at the interventions to postpone or reverse cognitive decline, including the FINGER study in our manuscript (page 17, lines 304-308):

“Another suggestion derivable from our findings is that older adults with hearing impairments might profit from interventions designed to prevent cognitive decline, such as physical activity [51] and the multidomain intervention (combination between diet, exercise, cognitive training and vascular risk monitoring) [52, 53]. Well-designed future studies are needed to determine whether those interventions can attenuate or prevent hearing impairment.”

Methodological issues:

- I agree with the methodological issues pointed out by both reviewers. I would like to ask you address them by carrying out and reporting the suggested additional analyses.

We have carried out and reported the suggested additional analyses:

1. We have included survey weight in our analyses.

2. We have performed a bivariate analyses to compare the analytic and excluded sample.

3. We have performed a correlation analysis between subjective hearing impairment and the objective hearing test at wave 7.

- It seems reasonable to select people without subjective hearing impairment at t1, but I suggest to include more information on the differences between this subjective measure (asking about hearing impairment with usual aids, i.e. compensating possibly existing hearing impairment) and an objective hearing test.

We have included the information on the differences between the analytic sample and excluded sample in Supplementary Table 1 and Methods section (page 6 lines 118-125):

“Differences between the analytic sample (3,615) and excluded sample (n=726) were tested with Kruskal-Wallis one-way analysis of variance for numerical variables and ordinal chi-square tests for categorical variables as appropriates (Supplementary Table 1). There were more women than men in the analytic sample, conversely in the analytic sample (p<0.001). The excluded sample has lower episodic memory scores on average (p<0.001), lower education attainment (p<0.001), poorer (p<0.001), and older (74 versus 72.2; p<0.001). The proportion of excluded sample with objective hearing impairment in wave 7 (26.8%) was higher than that of analytic sample (6.9).”

- What is the correlation between subjective hearing impairment at wave 7 and the objective hearing test?

We have performed the Spearman correlation analysis to identify the correlation between subjective hearing function and objective hearing tests at wave 7 (page 6, line 132-135):

“The objective hearing function (as measured by the number of tones that participants heard in both ears) at wave 7 has significant correlation with the subjective hearing function (in 5 Likert scale) at the same wave (Spearman correlation=-0.4768, p<0.001).”

- The claim to predict hearing impairment based on the cognitive trajectories more than a decade in advance needs to be substantiated by a test on the ability of the intercept of these trajectories, i.e. a time point more than a decade earlier than the hearing test, to predict hearing impairment.

As we have only the longitudinal information for 14 years, we have revised the words “over the preceding decade” into “during the preceding decade” (page 4 lines 89-91). 

 

Reviewer 1:

This article adds to the growing literature teasing out hearing-cognition relations in later adulthood. Unlike the majority of research in this area which focuses on hearing as a risk-factor for future cognitive decline – this study considers the opposite direction and explores cognitive decline as a predictor of future hearing impairment. This is its key contribution to the literature.

The article is clearly written, the research question is justified with good rationale, and the methods relatively sound. I have the following comments to make:

Introduction:

Page 3, line 68: As I understand it, Kiely et al did include time-dependent measure of cognition and found incidence of cognitive impairment to be linked with higher hearing thresholds. So, yet limitations of Kiely et al remain – they used a verbally administered screen of global cognitive function (the MMSE) and so were unable to look at cognitive domains or actuals trajectories. Thus, the work of this paper provides important addition.

One other important study to include is Armstrong et al 2018 Journals of Gerontology ( https://doi.org/10.1093/gerona/gly268) who used cross-lagged panel regression to explore reciprocal relationship between hearing and cognition over two time points. They did not find prior cognition to predict future hearing levels. I also encourage authors to consider the findings of Lindenberger and Ghisletta 2009 Psych and Aging (https://psycnet.apa.org/record/2009-03151-001) and the earlier work of Baltes and Lindenberger (‘94, ‘97) who laid the theoretical foundations for this research question.

Thank you for the comments. We have revised our description on Kiely and colleagues’ work in page 6 line 68, and included the suggested literature in our Introduction section (page 3-4, lines 67-84):

“In relation to the possibility that declining cognitive capacity leads to hearing loss, prior study hypothesise that the declining cognitive ability results in poorer auditory perception as reduced cognitive capacity may constrain perceptual processing of stimuli [12, 15, 16]. Alternatively, hearing loss and cognitive decline may be associated because of a shared common factor(s) [13]. 

A large amount of experimental research demonstrates that cognitive factors – including memory, attention and processing speed – are critical for ‘listening’ [17-20]. Cognitive factors also impact on performance on tests of low level auditory perception, including detection of pure tones [21]. But longitudinal evidence for declining cognitive capacity leading to hearing loss is limited. To our knowledge, only one study has shown that cognitive impairment was associated with faster declines in hearing threshold [22]. That study used data from two-waves of the Mini-Mental State Examination [23], a verbally administered screen of global cognitive function, to measure the cognitive ability. Unfortunately, the MMSE is sensitive to hearing function [24] and this susceptibility may have confounded the results. In contrast, a study using two waves of data from the Baltimore Longitudinal Study of Aging found that cognitive function in the first wave did not correlate with change in audiometrically assessed hearing function in the second wave [25]. However, those studies used data from two time points only 2 years apart which restrict potential to characterise trajectories of the cognitive function.”

Methods

A self-rated hearing item has been applied to select the sample. This is appropriate as it restricts the sample to those more likely to be later identified with hearing impairment due to age related processes.

The definition of audiometric defined hearing impairment used at wave 7 is also reasonable, but should be made clear that this differs from the more commonly applied PTA>25dB (mild), PTA>40dB (moderate) for 500-4000 Hz etc…

We have include the use of the HearCheck Screen in the limitation section (page 17, lines 316-322):

“Thirdly, the objective measures of hearing function involved the HearCheck audiometric screener device instead of full audiometry. The accuracy of HearCheck has been established in a sample of adults a similar age to those in the cohort in this study [55]. HearCheck had 89% sensitivity and 62% specificity with respect of audiometrically tested hearing at 35 dB HL.”

We have also included the reasons of using 35 dB HL at 1000 Hz and 3000 Hz as the threshold in this study (page 6, lines 133-136):

“These thresholds have been identified as being the most useful frequencies for screening for hearing impairment, indicating a level of hearing impairment such that a person would likely benefit from a hearing aid [27].”

Cognitive measures appear to be verbally administered – associations between hearing and cognition may therefore be (in part) an artefact of verbal assessment. This is a common limitation in this line of research but should be acknowledged in the discussion.

We have include the limitation of cognitive measures we used in this study in limitation section (pages 17-18, lines 320-322):

“Fourth, the cognitive ability was measured using verbally administered tests. The significant relationship between the trajectories of recall memory and hearing function in this study may therefore be an artefact of verbal assessment.”

Much more information on the modelling of the latent classes is needed. Are the latent classes derived from intercept and slope factors from a latent growth curve? How was variance within the classes estimated? etc.

We have included the explanation of the latent class trajectory models in the Method section (pages 7-8, lines 168-177):

“In the first phase, we built latent class trajectory models to identify trajectories of recall memory scores over 13 years. Latent class trajectory models is an extension form of finite mixture modelling, which describes the course of recall memory scores through a regression function using continuous latent growth factors [32]. We used a latent class trajectory model as it could classify respondents into distinct groups based on their response patterns so that respondents within a group are more similar than respondents between groups. The intercept represent the level of recall memory scores at baseline. The change in recall memory scores over time is accounted for by the linear and quadratic slopes of the growth factors. We calculated the posterior probabilities for each trajectory taking into account the respondents’ age (at baseline), sex, and educational level.”

Table 1

Without referring to deep into the results section it is not clear what is meant by “most advantaged”. What is the variable stratifying the sample in the columns? This should be made obvious in the table header.

We have revised the table header to follow the name of four trajectories identified in the manuscript: the highest, 2nd, 3rd, and lowest recall memory trajectories.

Please ensure consistent reporting to decimal places. (e.g. age SD reported as 7 rather than 7.0 for class 3 and class 4)

We have revised the tables and make sure that the decimal places in all numbers reported are consistent.

Table 1 refers to wealth. Is this wealth or income? In text it appears to be referred to as income.

We have revised the word “income” into “wealth” in Table 1.

Figure 1 and the latent classes

These are interesting, but appear to simply reflect differences in levels? Are there differences in the way these classes are changing over time?

Yes, we have explained in Results section (page 12 lines 215-217) regarding the slopes of the different trajectories:

“The highest recall memory trajectory took on a strongly curvilinear shape and showed cognitive maintenance after age 50, while the cognitive trajectory of respondents included in the lowest group showed a linear decline.”

Why does the line in figures 1 for most advantaged class stop around age 85 whereas other classes lines continue to age 90? It looks like the grades of cognitive advantaged trajectories are strongly tied with age. Is this true? I would like to see mention of this on page 9-10 (lines 180-184) where a description of characteristics of each class is proved. i.e. most advantaged class has higher income, more education, more social engagement, higher levels of exercise AND is also younger, which may explain all these differences?

We have provided the description of characteristics of each class in page 12 lines 219-221, including the possible effect of age:

“Respondents with the highest recall memory trajectory were likely to perform better on the memory test over time, younger, to be wealthier, and better educated than those with lower recall memory trajectories.”

I wonder if it would have been simpler, and just as instructive, to regress hearing impairment at wave 7 on latent factors for intercept and slope that are estimated from a latent growth curve for cognition.

We have included the explanation of the latent class trajectory models in the Method section that we used latent class trajectory model as it could identify respondents into distinct groups based on their response patterns so that respondents within a group are more similar than respondents between groups. 

We further performed logistic regression model to identify the relationships between recall memory trajectories and the odds of having hearing impairment (page 8, lines 187-189):

“In the third phase, a logistic regression of hearing impairment on recall memory trajectories while controlling for other determinants in wave 7 was explored. The previously developed four-class trajectories were maintained.”

Table 2

Am I correct in interpreting the coefficients as the log-odds as the outcome is binary? Is this effectively a logistic regression? Some readers might benefit from reporting of odds ratios.

We have revised the analyses and reporting the results using odds ratio (95% confidence intervals).

Discussion:

Overall the discussion is very nicely presented.

Again, I felt the manuscript would benefit from bringing stronger theoretical foundations earlier outlined by Baltes and Lindenberger. Schneider and Pichora-Fuller’s chapter in Craik and Salthouse’ Handbook of Aging and Cognition (2000) also give very thorough account of this.

Thank you for the comment. We are agree with it. We have thus added the information to compliment the potential mechanisms presented in Introduction section (Page 3-4 Lines 64-85):

A number of mechanisms are proposed in the literature: declining cognitive capacity leading to hearing loss, poorer hearing function causing accelerated cognitive decline [6-8], and the presence of third factors causing both types of decline [13, 14]. In relation to the possibility that declining cognitive capacity leads to hearing loss, prior study hypothesise that the declining cognitive ability results in poorer auditory perception as reduced cognitive capacity may constrain perceptual processing of stimuli [12, 15, 16]. Alternatively, hearing loss and cognitive decline may be associated because of a shared common factor(s) [13]. 

A large amount of experimental research demonstrates that cognitive factors – including memory, attention and processing speed – are critical for ‘listening’ [17-20]. Cognitive factors also impact on performance on tests of low level auditory perception, including detection of pure tones [21]. But longitudinal evidence for declining cognitive capacity leading to hearing loss is limited. To our knowledge, only one study has shown that cognitive impairment was associated with faster declines in hearing threshold [22]. That study used data from two-waves of the Mini-Mental State Examination [23], a verbally administered screen of global cognitive function, to measure the cognitive ability. Unfortunately, the MMSE is sensitive to hearing function [24] and this susceptibility may have confounded the results. In contrast, a study using two waves of data from the Baltimore Longitudinal Study of Aging found that cognitive function in the first wave did not correlate with change in audiometrically assessed hearing function in the second wave [25]. However, those studies used data from two time points only 2 years apart which restrict potential to characterise trajectories of the cognitive function.”

 We further explored those potential mechanisms and included the theoretical foundations in the Discussion section (page 15-16 lines 263-293):

“From a theoretical perspective, this study provides the first evidence that decreased recall memory ability predicts hearing impairment. The pathways between cognition and hearing function among older adults are not well understood. One plausible mechanism of the relationship is neurobiological. Research has established the positive correlations between cognitive function and total brain volume [34-36], which may influence the functioning of auditory processes in the brain. Histopathological involvement of auditory cortices has been found in Alzheimer’s disease and other dementias [37-39]. The general process in the central auditory system includes matching new inputs with existing memory as well as experience of the auditory world with other sensory systems [40, 41]. In addition to the process within the auditory system, there is a cortico-cortical process that provides integration of the auditory process and other sensory systems, memory, knowledge, and decision-making processes. The neurodegenerative process may disrupt both central auditory and cortico-cortical processes, leading to deficient perception of sounds and a more cautious or impaired decision-making process. A study in the US found that auditory cortical activity as measured by P50 amplitude and P300 latency decreased with normal ageing and decreased additionally with mild cognitive impairment [42]. 

Another potentially relevant mechanism is a shared underlying pathology, such as vascular diseases or intrinsic cellular ageing, that leads to ageing of both the brain and the auditory organs [13, 14, 16, 43, 44]. However, further explanation is warranted because the association between recall memory trajectories and the presence of hearing impairment in the present analyses remained after controlling for the effects of potential covariates (e.g. age, stroke, heart diseases).

Finally, recent trajectories of recall memory scores could also potentially predict hearing impairment through the broader influence of cognitive ability on physical activity. Likewise, social networks could mediate the observed association of cognition and a higher probability of hearing impairment. Better cognition may facilitate physical activity [45], which in turn protects older adults against chronic diseases [46, 47]. The presence of chronic diseases, including stroke, diabetes, hypertension and other cardiovascular diseases, are among the risk factors for hearing impairment [28-30]. Better cognitive function is associated with richer social networks [48], and similar to physical activity, social relationships are important for good health [49, 50].”

There is discussion of clinical perspectives and managing hearing loss. What is missing (or could at least be made more clearly) is consideration of the implications of these findings for hearing aid use. If cognitive impairment/decline/problems predict future hearing impairment, then this may presents a challenge for adjustment to new hearing aids and effective use of hearing aids among those with cognitive limitations.

We have include the implications of our findings for hearing aid use in page 17 lines 300-303:

“Our findings that cognitive decline predicts future hearing impairment may also presents a challenge for adjustment to new hearing aids and effective use of hearing aids among those with cognitive limitations.”

 

Reviewer 2:

This study evaluates associations between classes of recall memory trajectories and hearing impairment in old age. The classes were derived from latent class growth mixture models fitted to wave 1-7 from the English Longitudinal Study of Aging, and hearing impairment was defined as the inability to hear tones quieter than 35 dH HL at wave 7.

Given the way the data analyses were executed, I found it difficult to grasp what is the actual contribution of the study beyond what we already know from previous work, i.e., that memory performance and hearing impairment are correlated (at the population level) in old age. It is unclear what combination of parameters in the growth models are used to derive and define the classes, is it the level, slope, variance/covariance structure, residuals, and so forth? This information is essential for interpretation of the findings.

We have included the explanation of the latent class trajectory models in the Method section (pages 7-8, lines 168-177):

“In the first phase, we built latent class trajectory models to identify trajectories of recall memory scores over 13 years. Latent class trajectory models is an extension form of finite mixture modelling, which describes the course of recall memory scores through a regression function using continuous latent growth factors [32]. We used a latent class trajectory model as it could classify respondents into distinct groups based on their response patterns so that respondents within a group are more similar than respondents between groups. The intercept represent the level of recall memory scores at baseline. The change in recall memory scores over time is accounted for by the linear and quadratic slopes of the growth factors. We calculated the posterior probabilities for each trajectory taking into account the respondents’ age (at baseline), sex, and educational level.”

Also, reasons and justification for the model selection is unclear to my. What is actually gained by relying on latent class modelling in the context of the data and research questions? Is a multivariate normal assumption unrealistic for some reasons? Are there any neuronal factors or data patterns that may justify the mixture assumption? To me, it seems more informative to rely on the multivariate normal assumption and simply fit a conventional growth curve model to the data using the hearing impairment as covariate determining recall level and change, unless there are some specific reasons why that model could be misleading.

We used latent class trajectory model as it could identify respondents into distinct groups based on their response patterns so that respondents within a group are more similar than respondents between groups.

For matter of communicative precision I suggest that you replace “cognitive function” for “recall memory” in the title (and the rest of the manuscript).

We have replaced the words “cognitive function” with “recall memory” in the title and the rest of manuscript.

---

## [Decision Letter · Decision Letter 1]

24 Apr 2020

PONE-D-19-29591R1

Trajectories of recall memory as predictive of hearing impairment: A longitudinal cohort study

PLOS ONE

Dear Dr Maharani,

Thank you for submitting your manuscript to PLOS ONE. After careful consideration, we feel that it has merit but does not fully meet PLOS ONE’s publication criteria as it currently stands. Therefore, we invite you to submit a revised version of the manuscript that addresses the points raised during the review process.

We would appreciate receiving your revised manuscript by Jun 08 2020 11:59PM. To enhance the reproducibility of your results, we recommend that if applicable you deposit your laboratory protocols in protocols.io, where a protocol can be assigned its own identifier (DOI) such that it can be cited independently in the future. For instructions see: http://journals.plos.org/plosone/s/submission-guidelines#loc-laboratory-protocols

We look forward to receiving your revised manuscript.

Kind regards,

Bolajoko O. Olusanya, MBBS, FMCPaed, FRCPCH, PhD

Academic Editor

PLOS ONE

Reviewers' comments:

Reviewer's Responses to Questions

**Comments to the Author**

1. If the authors have adequately addressed your comments raised in a previous round of review and you feel that this manuscript is now acceptable for publication, you may indicate that here to bypass the “Comments to the Author” section, enter your conflict of interest statement in the “Confidential to Editor” section, and submit your "Accept" recommendation.

Reviewer #1: (No Response)

Reviewer #3: (No Response)

2. Is the manuscript technically sound, and do the data support the conclusions?

Reviewer #1: Yes

Reviewer #3: Yes

3. Has the statistical analysis been performed appropriately and rigorously? 

Reviewer #1: Yes

Reviewer #3: Yes

4. Have the authors made all data underlying the findings in their manuscript fully available?

Reviewer #1: Yes

Reviewer #3: Yes

5. Is the manuscript presented in an intelligible fashion and written in standard English?

Reviewer #1: Yes

Reviewer #3: Yes

6. Review Comments to the Author

Reviewer #1: I am satisfied the authors have responded reasonably to all my prior comments, and I thank them for taking on board my suggestions. I only have the following 3 minor suggestions to make:

1) Please check citation [22 – Kiely et al] which examined MMSE change over more than 2 waves of data (4 waves spanning 11 years).

2) Despite being generally sceptical about GMMs I am overall happy with the methods and results as presented – but there are still details about the GMM I would like to see in supplementary table, as this would help make clear how these were estimated. For example, how was the within-class variance modelled? Was variance fixed to be constant within classes, or was it free to vary within classes? A supplementary table showing the full parameterisation of the GMM would clarify this.

3) My queries about Figure 1 (why does line end at age 85 end for one class, but extend beyond this range for other classes?) and comments about strong age gradient by trajectory classes were not addressed. Even though these models are adjusted for linear effects of age, I still wonder about the extent to which the substantive finding is driven by age?

Reviewer #3: In ‘Trajectories of recall memory as predictive of hearing impairment: a longitudinal cohort study,’ Maharani and colleagues examine the relationship in later life between cognitive function and subsequent hearing loss. They find that individuals with lower performance on a recall memory task are significantly more likely to develop hearing loss. The manuscript is clearly written and the analysis appear carefully done. A large sample size offsets some weakness in the assessment of hearing loss, which was self reported at the study onset and based on the HearCheck screener at the final time point. The manuscript should be of interest to readers of PLOS ONE. The main significance of the work is that it provides insight into the directionality of the relationship between cognitive decline and HL (the former potentially leading to the later).

One aspect of the results that is clear from figure 1 but could be clarified more in the text, throughout the manuscript, is that different classes of memory recall trajectory differ more in their starting point or mean than in the rate of decline.

A second issue that does not appear to be discussed is that the proportion of individuals with objective hearing loss in the study sample appear rather low for this age group, at 6.9%. This could be related to the requirement for HL at 500 Hz, which typically occurs later.

Line 31. It seems important to clarify here that absence of hearing loss was ‘self reported’

Lines 34-36. Rather than describing model selection here, it might be more useful to sketch out differences in the identified trajectories including different starting points.

Line 49. Remove ‘the’ after ‘1989’.

Line 113. Do the authors have information on which individuals used hearing aids? It seems like they should be excluded if possible since the objective hearing screening was likely not performed with a hearing aid.

Line 160. Please clarify what how moderate and vigorous exercise were distinguished. Was there a different question for each?

Line 194. Please clarify that this was a second analysis.

Line 209. It would helpful to have something at the top of the table indicating that ‘Highest’, ‘2nd', etc. refers to the recall trajectory. A header may be missing over ‘Drinking regularly’ since this is not a ‘Smoking Behavior’.

Line 215. Here is where it would be good to describe the difference in the intercepts/starting points.

Line 217. The decline does not appear ‘linear’ in Fig. 1.

Line 238. It would help to indicate the difference between the two columns at the top of the table, i.e., to somehow indicate that the right column includes the recall trajectory. Also, the meaning of the symbols (presumably related to significance) appears to be missing. Finally, the confidence interval for ‘vigorous physical activity’ contains 1 but is marked as significant. Please clarify.

Line 255. Noting the different intercepts for the various trajectories would be helpful here. Again, the decline does not appear ‘linear’.

7. PLOS authors have the option to publish the peer review history of their article (what does this mean?). If published, this will include your full peer review and any attached files.

Reviewer #1: No

Reviewer #3: No

---

## [Author Response · Author response to Decision Letter 1]

25 May 2020

Dear Reviewer,

We are very pleased with the review and your efforts in giving feedback on the manuscript. Your recommendations were very useful to strengthen the article. Below you will find your comments numbered and in italic, each followed by a short description of how the comment is implemented in the revised manuscript (in normal font). We have highlighted the changes within the revised manuscripts using coloured text. The number of page and paragraph in our responses refer to the revised manuscripts. We hope that we succeeded in strengthening the paper.

Yours sincerely,

The authors

Reviewer 1:

I am satisfied the authors have responded reasonably to all my prior comments, and I thank them for taking on board my suggestions. I only have the following 3 minor suggestions to make:

1) Please check citation [22 – Kiely et al] which examined MMSE change over more than 2 waves of data (4 waves spanning 11 years).

We have revised the citation in the Page 4 lines 79:

‘To our knowledge, only one study has shown that cognitive impairment was associated with faster declines in hearing threshold [22]. That study used data from four waves spanning 11 years of the Mini-Mental State Examination, a verbally administered screen of global cognitive function, to measure the cognitive ability [22].’

2) Despite being generally sceptical about GMMs I am overall happy with the methods and results as presented – but there are still details about the GMM I would like to see in supplementary table, as this would help make clear how these were estimated. For example, how was the within-class variance modelled? Was variance fixed to be constant within classes, or was it free to vary within classes? A supplementary table showing the full parameterisation of the GMM would clarify this.

In our analysis, we performed latent class growth analysis, in which the variance and covariance estimates for the growth factors within each class are assumed to be fixed to zero. By this assumption, all respondents’ growth trajectories within a class are homogenous. We provide the regression parameters of the latent class models in Supplementary Table 2.

3) My queries about Figure 1 (why does line end at age 85 end for one class, but extend beyond this range for other classes?) and comments about strong age gradient by trajectory classes were not addressed. Even though these models are adjusted for linear effects of age, I still wonder about the extent to which the substantive finding is driven by age?

We have added the explanation on Figure 1 in Page 12 lines 220-222:

‘The trajectory of recall memory among the respondents in the highest recall memory group ends at 86 as they are younger than other age groups. The oldest respondent in that group was aged 86 in the final wave.’

 

Reviewer 3

In ‘Trajectories of recall memory as predictive of hearing impairment: a longitudinal cohort study,’ Maharani and colleagues examine the relationship in later life between cognitive function and subsequent hearing loss. They find that individuals with lower performance on a recall memory task are significantly more likely to develop hearing loss. The manuscript is clearly written and the analysis appear carefully done. A large sample size offsets some weakness in the assessment of hearing loss, which was self-reported at the study onset and based on the HearCheck screener at the final time point. The manuscript should be of interest to readers of PLOS ONE. The main significance of the work is that it provides insight into the directionality of the relationship between cognitive decline and HL (the former potentially leading to the later).

1. One aspect of the results that is clear from figure 1 but could be clarified more in the text, throughout the manuscript, is that different classes of memory recall trajectory differ more in their starting point or mean than in the rate of decline.

We have added the explanation on Figure 1 in Page 12 lines 219-220:

‘The recall memory trajectories in Figure 1 differ in the intercepts and mean of the memory scores.’

2. A second issue that does not appear to be discussed is that the proportion of individuals with objective hearing loss in the study sample appear rather low for this age group, at 6.9%. This could be related to the requirement for HL at 500 Hz, which typically occurs later.

We have added the discussion on the low prevalence of objective hearing impairment in Page 17-18 lines 320-329:

‘The proportion of the respondents with objective hearing impairment included in this study was lower for the age group (6.9%). Prior studies using national samples showed that the prevalence of hearing impairment among adults aged 50 years and older ranges from 20 to 40% [55]. The definition of hearing loss in this study was not the usual four frequency average of thresholds at 500, 1000, 2000 and 4000 Hz. It was performance on a pure tone audiometry screen with hearing loss equivalent to better ear threshold >35 dB HL at 1000 or 3000 Hz. Hearing loss usually appears first at higher frequencies, so including lower frequencies in an average hearing loss may make the hearing loss index less sensitive to early hearing loss. The 3000 Hz criterion for our hearing loss measure should make it reasonably sensitive to early hearing loss.’

3. Line 31. It seems important to clarify here that absence of hearing loss was ‘self-reported’

We have revised the sentence in Page 2 line 31 into:

‘The study population consisted of 3,615 adults aged 50+ who participated in the first wave of the English Longitudinal Study of Ageing, who had no self-reported hearing impairment in Wave 1, and who underwent a hearing test in Wave 7.’

4. Lines 34-36. Rather than describing model selection here, it might be more useful to sketch out differences in the identified trajectories including different starting points.

We have added the explanation in the Page 2 lines 37-38:

‘The different recall memory trajectories represent different starting points and mean of the memory scores.’

5. Line 49. Remove ‘the’ after ‘1989’.

We have removed the word ‘the’ after ‘1989’ in Page 3 line 50.

6. Line 113. Do the authors have information on which individuals used hearing aids? It seems like they should be excluded if possible since the objective hearing screening was likely not performed with a hearing aid.

We are interested in hearing impairment (pathology) indexed by pure tone audiometry test, not functional hearing. There is thus no basis for removing hearing aid users in our study. Excluding hearing aid users would exclude a high proportion of people with audiometric hearing loss, and reduce sensitivity of the analysis. 

7. Line 160. Please clarify what how moderate and vigorous exercise were distinguished. Was there a different question for each?

We have included the explanation on Page 7 lines 162-164:

‘We categorised respondents as engaging in moderate and vigorous physical exercise if they reported exercising as least once a week. The respondents responded to the questions whether they take part in sports or activities that are vigorous and moderately energetic, respectively.’

8. Line 194. Please clarify that this was a second analysis.

We have clarified in Page 9 line 196 that the analysis was the second analysis.

9. Line 209. It would helpful to have something at the top of the table indicating that ‘Highest’, ‘2nd', etc. refers to the recall trajectory. A header may be missing over ‘Drinking regularly’ since this is not a ‘Smoking Behavior’.

We have revised the top of Table 1 and included ‘recall memory’ to better explain the recall trajectory. We have also included the header ‘Drinking behaviour’ in the Table.

10. Line 215. Here is where it would be good to describe the difference in the intercepts/starting points.

We have added the explanation about the difference in intercepts in the Page 2 lines 37-38:

‘The different recall memory trajectories represent different intercepts and mean of the memory scores.’

11. Line 217. The decline does not appear ‘linear’ in Fig. 1.

We have removed the sentence in Page 12 line 219.

12. Line 238. It would help to indicate the difference between the two columns at the top of the table, i.e., to somehow indicate that the right column includes the recall trajectory. Also, the meaning of the symbols (presumably related to significance) appears to be missing. Finally, the confidence interval for ‘vigorous physical activity’ contains 1 but is marked as significant. Please clarify.

We have included the information on the different models presented in Table 2, added the information of the symbols and revised the significance for ‘vigorous physical activity’.

13. Line 255. Noting the different intercepts for the various trajectories would be helpful here. Again, the decline does not appear ‘linear’.

We have rewritten the sentence in Page 15 lines 257-259:

‘Supporting prior research [33], we identified the highest recall memory trajectory and the lowest recall memory trajectory over time.’

---

## [Editor Report · Decision Letter 2]

1 Jun 2020

Trajectories of recall memory as predictive of hearing impairment: A longitudinal cohort study

PONE-D-19-29591R2

Dear Dr. Maharani,

We are pleased to inform you that your manuscript has been judged scientifically suitable for publication and will be formally accepted for publication once it complies with all outstanding technical requirements.

With kind regards,

Bolajoko O. Olusanya, MBBS, FMCPaed, FRCPCH, PhD

Academic Editor

PLOS ONE
---

## [Editor Report · Acceptance letter]

9 Jun 2020

PONE-D-19-29591R2 

Trajectories of recall memory as predictive of hearing impairment: A longitudinal cohort study 

Dear Dr. Maharani:

I'm pleased to inform you that your manuscript has been deemed suitable for publication in PLOS ONE. Congratulations! Your manuscript is now with our production department. 

Kind regards, 

on behalf of

Dr. Bolajoko O. Olusanya 

Academic Editor

PLOS ONE